# RIM is essential for stimulated but not spontaneous somatodendritic dopamine release in the midbrain

Brooks G Robinson[1]*, Xintong Cai[2], Jiexin Wang[2], James R Bunzow[1], John T Williams[1], Pascal S Kaeser[2]*

[1]The Vollum Institute, Oregon Health & Science University, Portland, United States; [2]Department of Neurobiology, Harvard Medical School, Boston, United States

**Abstract** Action potentials trigger neurotransmitter release at active zones, specialized release sites in axons. Many neurons also secrete neurotransmitters or neuromodulators from their somata and dendrites. However, it is unclear whether somatodendritic release employs specialized sites for release, and the molecular machinery for somatodendritic release is not understood. Here, we identify an essential role for the active zone protein RIM in stimulated somatodendritic dopamine release in the midbrain. In mice in which RIMs are selectively removed from dopamine neurons, action potentials failed to evoke significant somatodendritic release detected via D2 receptor-mediated currents. Compellingly, spontaneous dopamine release was normal upon RIM knockout. Dopamine neuron morphology, excitability, and dopamine release evoked by amphetamine, which reverses dopamine transporters, were also unaffected. We conclude that somatodendritic release employs molecular scaffolds to establish secretory sites for rapid dopamine signaling during firing. In contrast, basal release that is independent of action potential firing does not require RIM.
DOI: https://doi.org/10.7554/eLife.47972.001

*For correspondence:
robinbro@ohsu.edu (BGR);
kaeser@hms.harvard.edu (PSK)

Competing interests: The authors declare that no competing interests exist.

## Introduction

In addition to secretion from axonal nerve terminals, many neurons release neurotransmitters or neuromodulators from their somata and dendrites (*Ludwig et al., 2016*). Important examples include neuropeptides, monoamines and neurotrophins, and signaling through these pathways is essential for brain function. However, the somatodendritic secretory machinery is not well understood.

A prominent example for somatodendritic secretion is the release of dopamine in the ventral midbrain. Subsequent activation of dopamine receptors is important for regulating neuronal excitability, for the response to drugs of abuse, and for the control of motor function (*Bjijou et al., 1996*; *Crocker, 1997*; *Ford, 2014*; *Ludwig et al., 2016*; *Vezina, 1996*). Somatodendritic dopamine release is mediated by vesicular exocytosis, as it is sensitive to clostridial toxins (*Bergquist et al., 2002*; *Fortin et al., 2006*). Despite years of study, essential molecular machinery for somatodendritic dopamine release has not been identified, but specific SNARE requirements and high calcium sensitivity have been proposed (*Chen et al., 2011*; *Mendez et al., 2011*; *Witkovsky et al., 2009*).

Electrophysiological recordings from midbrain dopamine neurons revealed that somatodendritic dopamine release evokes a D2 receptor mediated inhibitory postsynaptic current (D2-IPSC) that is mediated by GIRK channels and rises in 200 ms and decays in 500 ms (*Beckstead et al., 2004*). The D2-IPSC is produced by a high concentration of dopamine (100 µM; *Courtney and Ford, 2014*), and the duration is defined by efficient dopamine re-uptake through the dopamine transporter (DAT) (*Ford et al., 2009*). In addition, spontaneous dopamine release occurs and produces D2-IPSCs (*Gantz et al., 2013*). These studies indicate that somatodendritic release of dopamine can lead to

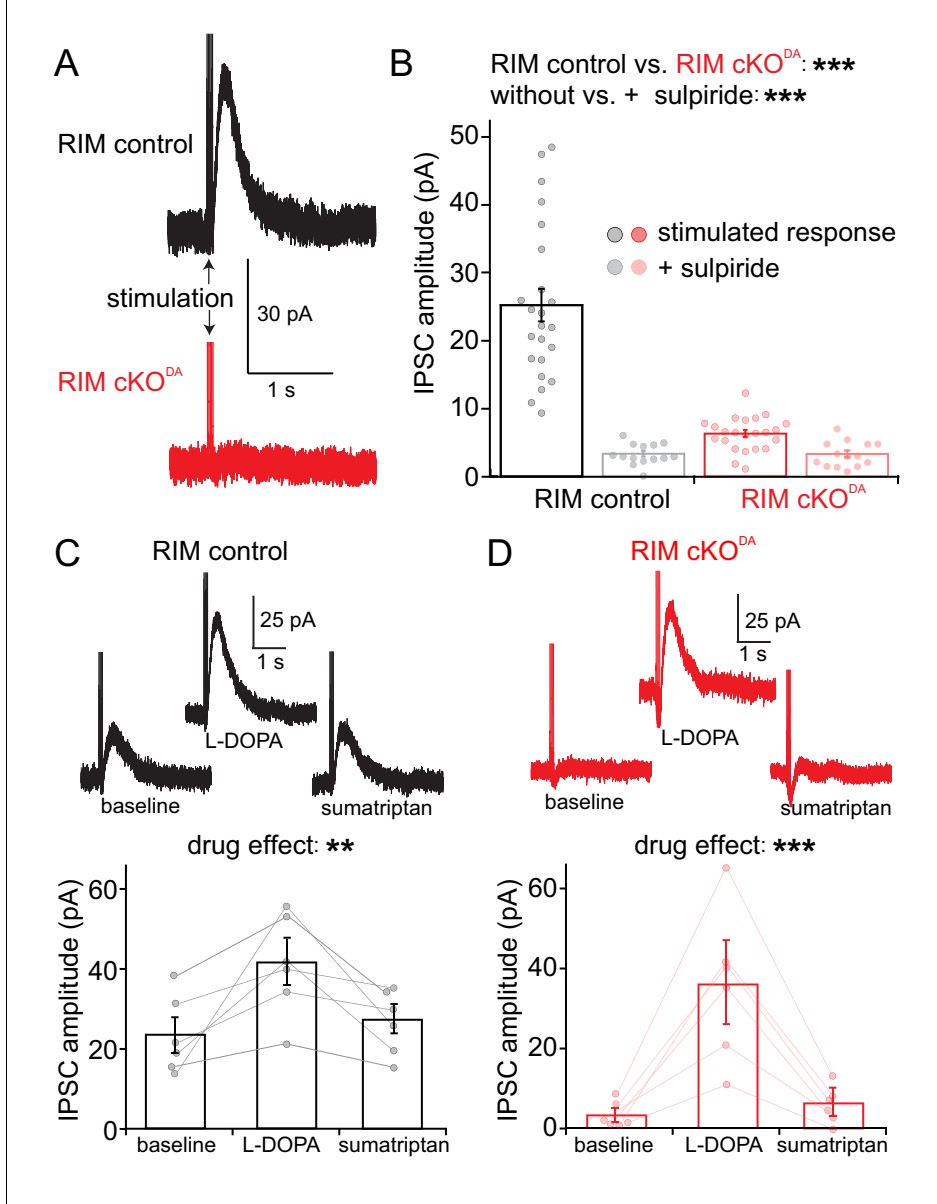

**Figure 1.** RIM is essential for stimulated somatodendritic dopamine release. Somatodendritic release was characterized in substantia nigra dopamine neurons. Release was induced using a monopolar electrode and measured by recording D2 receptor IPSCs in control mice (RIM control) and in mice with conditional knockout of RIM specifically in dopamine neurons (RIM cKO$^{DA}$). (A, B) Example traces (A) and quantification (B) of IPSCs in RIM control and RIM cKO$^{DA}$ mice with vs. without the presence of the D2 receptor antagonist sulpiride, n = 23 cells/6 mice in RIM control, and n = 23/6 in RIM cKO$^{DA}$, significance was calculated by two-way ANOVA and is reported in panel B (RIM control vs. RIM cKO$^{DA}$: $F_{(1)}$ = 57.63, p < 0.001; stimulated response vs. + sulpiride: $F_{(1)}$ = 60.71, p < 0.001), and was followed by Bonferroni post-hoc analysis (RIM control stimulated response vs. RIM control + sulpiride t = 9.70, p < 0.05; RIM cKO$^{DA}$ stimulated response vs. RIM cKO$^{DA}$ + sulpiride: t = 1.33, p > 0.05; RIM control stimulated response vs. RIM cKO$^{DA}$ stimulated response: t = 9.62, p < 0.05). (C, D) Example traces (top) and quantification (bottom) of IPSCs stimulated in RIM control slices (C) or RIM cKO$^{DA}$ slices (D) before and after treatment with L-DOPA (10 µM) and subsequent application of sumatriptan (1 µM, to inhibit dopamine release from serotonin terminals), n = 6 cells/6 mice in each group, significance was calculated by repeated measures ANOVA and is reported in panels C and D, (C: F = 10.44, p = 0.01 D: F = 22.75, p < 0.005), and was followed by Tukey's multiple comparison test (C: baseline vs. L-DOPA p < 0.05, L-DOPA vs. sumatriptan p < 0.05; D: baseline vs. L-DOPA p < 0.05, L-DOPA vs. sumatriptan p < 0.05). Data in B-D are shown as mean ± standard error of mean (SEM) and small circles represent individual cells.

*Figure 1 continued on next page*

*Figure 1 continued*

DOI: https://doi.org/10.7554/eLife.47972.002

The following figure supplements are available for figure 1:

**Figure supplement 1.** Comparison of dopamine neuron properties in RIM control and RIM cKO[DA] mice.
DOI: https://doi.org/10.7554/eLife.47972.003

**Figure supplement 2.** Cocaine enhances the IPSC amplitude and prolongs the IPSC decay.
DOI: https://doi.org/10.7554/eLife.47972.004

**Figure supplement 3.** Sensitivity of D2 receptors is not altered by RIM removal.
DOI: https://doi.org/10.7554/eLife.47972.005

activation of nearby receptors. Hence, mechanisms for targeting somatodendritic secretion towards specific membrane domains close to receptor clusters on target cells must be present.

The goal of this study was to identify molecular machinery that could provide for precise targeting of somatodendritic secretion of dopamine, focusing on the active zone organizer RIM. RIM localization within neurons is thought to be restricted to axons, where it is present in small clusters within active zones and organizes these release sites (*de Jong et al., 2018*; *Kaeser et al., 2011*; *Tang et al., 2016*; *Wang et al., 2016*; *Wang et al., 1997*). Recent reports suggest that specialized RIM isoforms may localize and function in dendrites (*Alvarez-Baron et al., 2013*), and postsynaptic roles for RIM1α in LTP have been proposed (*Wang et al., 2018*). In dopamine neurons, RIM was recently identified to be localized to active-zone like secretory sites in striatal dopamine axons, and RIM is essential for dopamine secretion in the striatum (*Liu et al., 2018*). It is not known whether RIM is present in dopamine neuron somata and dendrites.

## Results and discussion

We generated RIM cKO[DA] mice, in which RIM1 and RIM2 were specifically removed in dopamine neurons by crossing conditional RIM 'floxed' alleles to DAT[IRES-cre] mice (*Bäckman et al., 2006*; *Liu et al., 2018*). Mice heterozygous for the RIM floxed and DAT[IRES-cre] alleles were used as controls (RIM control). We prepared acute brain slices from these mice and recorded from individual substantia nigra dopamine neurons. There were no differences in cell capacitance, resistance, spontaneous firing rates, $I_h$ current, or the current measured immediately following break-in between RIM cKO[DA] and RIM control mice (*Figure 1—figure supplement 1*). Electrical stimulation (5 pulses at 40 Hz) was used to elicit dopamine release measured as D2-IPSCs. If an IPSC was not initially present, the stimulation intensity was increased and/or the electrode was relocated. In RIM controls, IPSCs of 10 to 50 pA were reliably induced (*Figure 1A and B*), and the time course of dopamine transmission was dependent on reuptake rather than diffusion of dopamine in these control animals (*Figure 1—figure supplement 2*). Compellingly, D2-IPSCs were difficult or impossible to evoke in RIM cKO[DA] slices (*Figure 1A and B*). The currents in most recordings were indistinguishable from baseline, and only one stimulated response was over 10 pA. The stimulating electrode was relocated 21 times in RIM cKO[DA], while only six relocations were necessary in RIM control, illustrating the strong impairment in RIM cKO[DA] mice. We conclude that somatodendritic dopamine release strongly depends on RIM, suggesting that it is controlled by active zone-like scaffolds.

It is possible that the secretory defect in RIM cKO[DA] slices is misjudged because RIM cKO[DA] may lead to altered expression or function of D2 receptors. Application of the dopamine precursor L-DOPA increases D2-IPSCs (*Beckstead et al., 2004*; *Gantz et al., 2015*), in part due to higher levels of vesicular dopamine in dopamine neurons. A significant component, however, results from the metabolism of L-DOPA in serotonin terminals, which then release dopamine and enhance D2-IPSCs. This component is blocked by the serotonin autoreceptor agonist sumatriptan (*Gantz et al., 2015*). This phenomenon was leveraged here. In RIM control slices, L-DOPA increased D2-IPSCs, and sumatriptan partially reversed it (*Figure 1C*). In RIM cKO[DA] slices, initial stimulation did not evoke significant D2-IPSCs, but application of L-DOPA resulted in sizeable IPSCs that were blocked by sumatriptan (*Figure 1D*). Hence, D2 receptors are present and functional in RIM cKO[DA] mice, and D2-IPSCs are produced when dopamine is artificially released from sources other than the dopamine neurons. An alternative approach to assess D2 receptors was to superfuse dopamine onto the slices (1 and 100 μM) and to determine the current density. There was no significant difference between

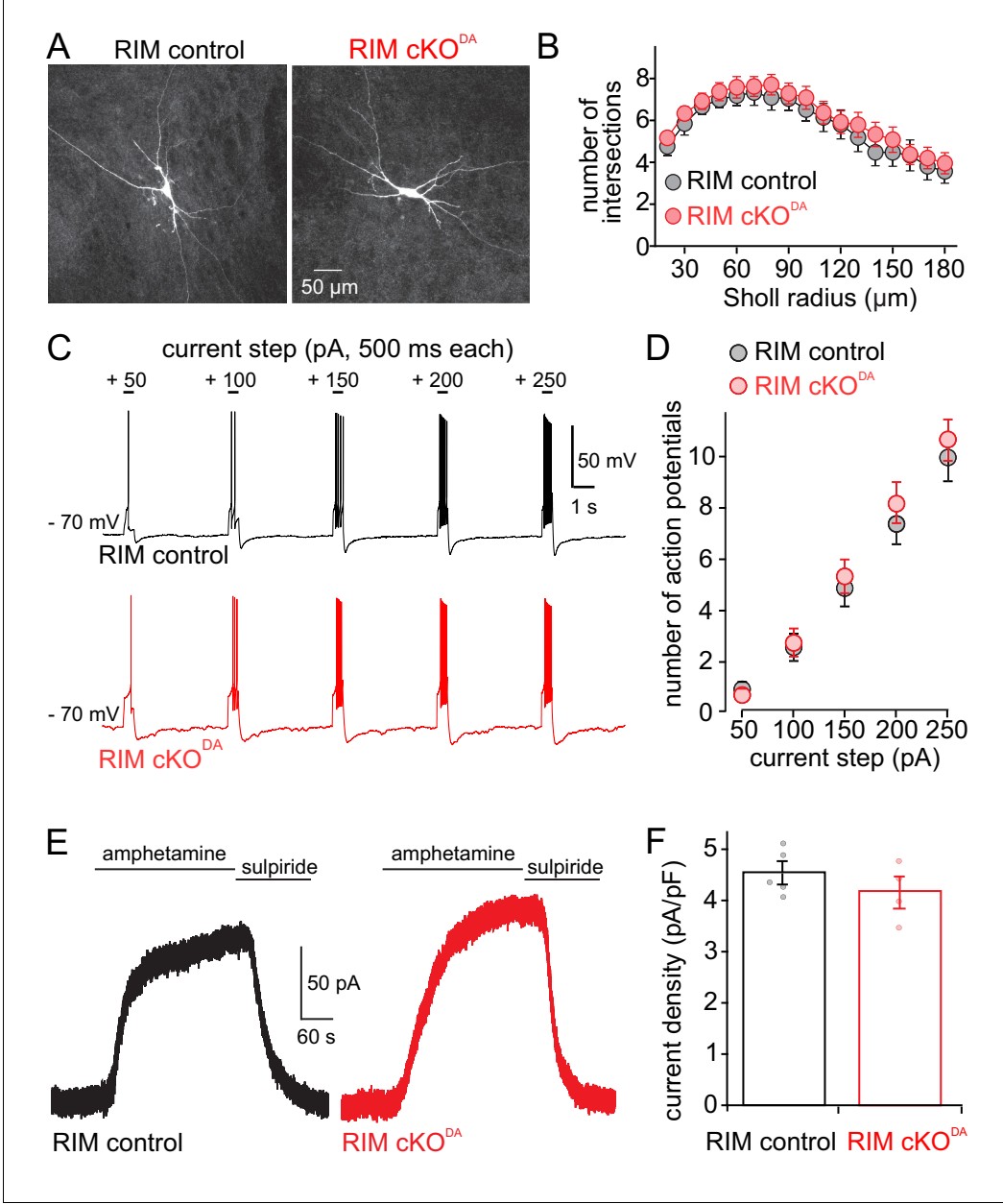

**Figure 2.** Dopamine neuron shape, excitability and amphetamine induced dopamine release are unaffected by RIM knockout. (**A, B**) Example images (**A**) and Sholl analysis (**B**) of individual neurobiotin-filled dopamine neurons in RIM control (n = 21 cells/6 animals) and RIM cKO[DA] (n = 24/6) slices. RIM control and RIM cKO[DA] neurons in B were compared using two-way ANOVA (F (1, 43) = 0.53, p = 0.47). (**C, D**) Neuron excitability was tested by hyperpolarizing cells to − 70 mV and applying progressively larger 500 ms long positive current steps. Example traces (**C**) and quantification (**D**) of action potential firing recorded in current clamp are shown. The number of action potentials during each step was quantified in RIM control (n = 27/6) and RIM cKO[DA] (n = 27/6), and then compared using two-way ANOVA (current step size effect F(4, 240) = 71.50, p < 0.0001; RIM control vs. RIM cKO[DA] F(1, 240) = 0.85, p = 0.36). (**E, F**) Example traces (**E**) and quantification of current density (**F**) from RIM control and RIM cKO[DA] mice of D2 receptor currents produced by bath application of amphetamine (10 or 30 μM, which causes the reverse transport of dopamine into the extracellular space); n = 5/5 in RIM control, 4/4 in RIM cKO[DA], compared by Student's t-test (t = 1.12, p = 0.30). Data in B, D and F are shown as mean ± SEM and small circles in F represent individual cells.

DOI: https://doi.org/10.7554/eLife.47972.006

The following figure supplement is available for figure 2:

*Figure 2 continued*

**Figure supplement 1.** The amphetamine response is not saturated.
DOI: https://doi.org/10.7554/eLife.47972.007

RIM control and RIM cKO$^{DA}$ animals (*Figure 1—figure supplement 3*). We conclude that D2 receptor levels and function were not strongly altered upon RIM knockout.

RIM cKO$^{DA}$ may have resulted in altered dopamine neuron structure or excitability, or in a loss of dopamine vesicles in the somatodendritic compartment. To assess these alternative explanations for the loss of dopamine release, we first characterized the morphology of dopamine neurons that were dye-filled through a patch pipette, fixed, and imaged by confocal microscopy. RIM control and RIM cKO$^{DA}$ neurons were indistinguishable in shape as assessed by Sholl analyses (*Figure 2A and B*). We next characterized excitability of the neurons in acute brain slices. We injected depolarizing currents of increasing size into individual neurons, and measured the resulting number of action potentials in those neurons. On average, RIM cKO$^{DA}$ neurons fired the same number of action potentials in response to these currents (*Figure 2C and D*). Finally, amphetamine was used to reverse the plasma membrane and vesicular dopamine transporters (DAT and vMAT2, respectively). The resulting increase in extracellular dopamine triggered D2-GIRK currents that were similar in RIM cKO$^{DA}$ and RIM control slices (*Figure 2E and F*), indicating that the somatodendritic level of dopamine containing vesicles is similar between RIM cKO$^{DA}$ and RIM control mice. Hence, RIM removal did not strongly alter the size and development of dopamine neurons.

Basal or spontaneous release of dopamine occurs in vivo and in slices, and its detection as D2-IPSCs is facilitated in a low concentration of cocaine and forskolin (*Gantz et al., 2013*). Spontaneous D2-IPSCs were readily detected in RIM control and RIM cKO$^{DA}$ slices (*Figure 3A and B*) and were blocked by the D2 receptor antagonist sulpiride. They were identical in amplitude and 20% peak width in both conditions, and had a non-significant trend towards increased frequency in RIM cKO$^{DA}$ mice (*Figure 3C-E*). We conclude that, while removal of RIM abolishes stimulated IPSCs, spontaneous release does not necessitate RIM. The dichotomy between evoked and spontaneous somatodendritic dopamine release is surprising, because at typical synapses, for example glutamatergic or GABAergic synapses in the hippocampus, RIM is important for both forms of release (*Deng et al., 2011*; *Kaeser et al., 2011*). The normal amplitudes and kinetics of spontaneous dopamine release further strengthen the point that D2 receptor function and localization are not altered in RIM cKO$^{DA}$ slices.

RIM is important for rapid and precise exocytosis of synaptic vesicles (*Kaeser et al., 2011*; *Koushika et al., 2001*; *Müller et al., 2012*; *Wang et al., 2016*). Dopamine has long been considered a neuromodulator that impacts circuits in a paracrine fashion. However, it was recently found that axonal dopamine release in the striatum is fast and depends on active zone-like release sites (*Liu et al., 2018*). Here, we establish that RIM is essential for stimulated release from dopamine neuron somata and dendrites that is detected via D2-IPSCs. Three points suggest that RIM organizes or generates secretory sites to mediate somatodendritic release. First, it is likely that RIM function is similar at somatodendritic and axonal release sites. At axonal sites, RIM controls the docking and priming of vesicles (*Deng et al., 2011*), the close by tethering of Ca$^{2+}$ channels (*Han et al., 2011*; *Kaeser et al., 2011*), and the coupling of these functions to PI(4,5)P$_2$ (*de Jong et al., 2018*), a target membrane phospholipid that is important for synaptic vesicle release (*Milosevic et al., 2005*; *Di Paolo et al., 2004*). Second, somatodendritic dopamine release has a high initial release probability (*Beckstead et al., 2007*), indicating that scaffolding mechanisms to tether dopamine-laden vesicles to Ca$^{2+}$ channels and other secretory machinery are essential. Third, although the overall time course of the dopamine IPSC is dependent on GPCR signaling, IPSC activation requires a rapid rise in dopamine concentration (*Courtney and Ford, 2014*; *Ford et al., 2009*). This is best achieved by synchronous release of dopamine from nearby point sources. Together, these points suggest that RIM operates as a molecular scaffold to establish somatodendritic release sites that direct release towards D2 receptors on target cells. While it remains uncertain whether the time scales of dopamine coding require molecular machinery for rapid dopamine transmission, recent findings suggest that dopamine mediates effects on relatively fast time scales (*Howe and Dombeck, 2016*; *Menegas et al., 2018*; *da Silva et al., 2018*; *Yagishita et al., 2014*), and the machinery we identify here and in a previous study (*Liu et al., 2018*) could serve such roles. Regardless of the timing of

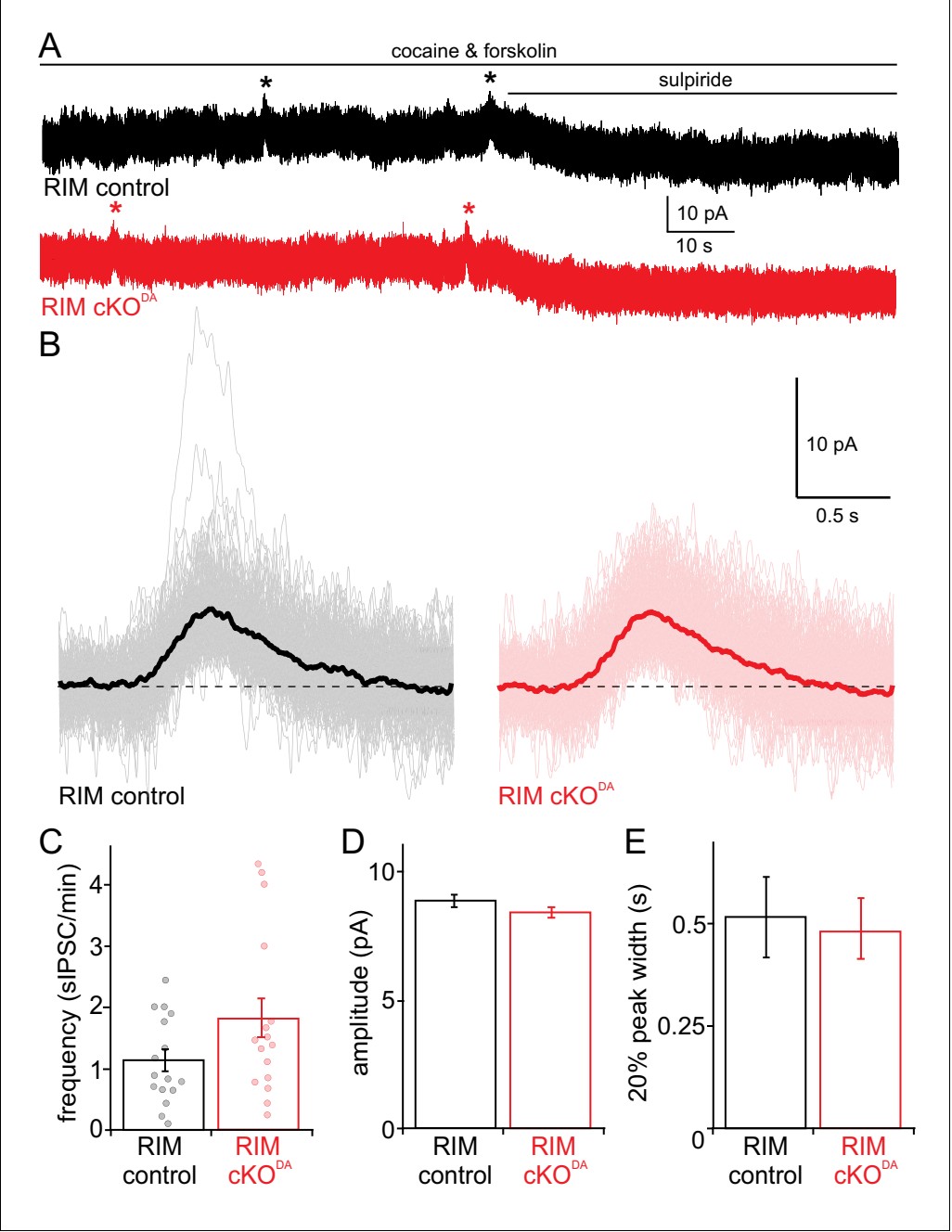

**Figure 3.** Spontaneous dopamine release is unaffected by RIM knockout. (**A, B**) Recording of spontaneous D2 receptor IPSCs (sIPSC) in RIM control and RIM cKO[DA] mice. Example traces (**A**) and aligned events (**B**, averages in bold) of D2 receptor IPSCs produced by spontaneous dopamine release from RIM control and RIM cKO[DA] slices in the presence of cocaine (300 nM) and forskolin (1 μM) followed by addition of the D2 receptor antagonist sulpiride (RIM control n = 161 events/16 cells/6 mice, RIM cKO[DA] n = 185/16/6). (**C**) The frequency of spontaneous IPSCs was quantified per cell in RIM control (n = 16 cells/6 mice) and RIM cKO[DA] (n = 16/6) slices, and the groups were compared by Student's t-test (t = 1.79, p = 0.09). (**D, E**) The amplitude (**D**) and 20% peak width (**E**) of the spontaneous IPSCs were analyzed and compared between all events of RIM control and RIM cKO[DA] mice using Student's t-test (n as in B; amplitude: t = 1.32, p = 0.19; half peak width: t = 1.46, p = 0.14). Data in C-E are shown as mean ± SEM and small circles in C represent individual cells.

DOI: https://doi.org/10.7554/eLife.47972.008

dopamine coding, this machinery is well suited to provide the high dopamine concentrations that are necessary to activate D2 receptors and for the regulation of dopamine signaling.

Compellingly, spontaneous dopamine release in the midbrain remains intact upon RIM knockout, indicating that molecular scaffolding mediated by RIM is dispensable for spontaneous dopamine release. Consistent with this point, spontaneous dopamine release is less dependent on $Ca^{2+}$-entry compared to stimulated release (*Gantz et al., 2013*). Together, these findings suggest that at least two modes of dopamine transmission exist. A large, fast, and directed mode, likely dependent on excitatory inputs and depolarization-induced $Ca^{2+}$ entry, requires RIM. A basal mode, in contrast, does not rely on RIM. It is unclear whether these two modes occur at the same or different locations. It is further possible that not all dopamine release is reported by D2-IPSCs, and that additional transmission modes exist.

In dopamine axons, RIM was only present in ~1/3 of the varicosities, and this fraction is likely responsible for stimulated dopamine release (*Liu et al., 2018*; *Pereira et al., 2016*). Whether a similar pattern is present in the somatodendritic compartment is unclear, but our findings raise the possibility that stimulated and spontaneous release occur at different locations in axonal and somatodendritic compartments. Given that the spontaneous dopamine IPSCs scale to stimulated IPSCs (*Gantz et al., 2013*), it is likely that the pre- and postsynaptic elements of somatodendritic transmission are in close proximity for both release modes. RIMs' scaffolding functions may allow for the synchronization of release from multiple vesicles at an individual site, from multiple sites, or from multiple neurons when activated simultaneously, allowing for more powerful activation of D2 receptors than spontaneous events.

Finally, our results establish important roles for RIM mediating rapid and efficient secretion from somata and dendrites, suggesting that active zone scaffolding is employed for fast exocytosis beyond that of synaptic vesicles in axonal boutons.

# Materials and methods

## Key resources table

| Reagent type (species) or resource | Designation | Source or reference | Identifiers | Additional information |
|---|---|---|---|---|
| Genetic reagent (mouse) | B6.SJL-Slc6a3[tm1.1(cre)Bkmn]/J | *Bäckman et al., 2006* | RRID:IMSR_JAX:006660 | |
| Genetic reagent (mouse) | Rims1[tm3Sud]/J | *Kaeser et al., 2008* | RRID:IMSR_JAX:015832 | |
| Genetic reagent (mouse) | Rims2[tm1.1Sud]/J | *Kaeser et al., 2011* | RRID:IMSR_JAX:015833 | |

## Animals

RIM cKO[DA] and RIM control mice were generated as described in *Liu et al. (2018)*, crossing mice with essential exons flanked by loxp sites in the *Rims1* ('floxed RIM1') and *Rims2* ('floxed RIM2') genes (*Kaeser et al., 2008*; *Kaeser et al., 2011*) with DAT[IRES-cre] mice (*Bäckman et al., 2006*). RIM cKO[DA] mice are homozogyote floxed for RIM1 and RIM2 and are heterozygous for DAT[IRES-cre], and RIM control mice are heterozygous for floxed RIM1, RIM2 and DAT[IRES-cre]. Male and female adult mice (100–160 days old) were used for all experiments, and mice were either littermates from the same litter or age-matched controls from the same intercrosses. All procedures and experiments were approved by and done in accordance with the policies of the IACUC at Oregon Health and Science University and at Harvard University.

## Slice preparation and electrophysiological recordings

Mice were deeply anesthetized with isoflurane and decapitated. Brains were rapidly removed and slices (222 μM thick) containing the substantia nigra were taken in the horizontal plane using a vibrating blade microtome and placed in a recovery chamber for > 30 min prior to experimentation. All preparation was done in warm (32–34° C) Krebs buffer containing (in mM) 126 NaCl, 1.2 $MgCl_2$, 2.4 $CaCl_2$, 1.4 $NaH_2PO_4$, 25 $NaHCO_3$, 11 D-glucose, along with 10 μM MK-801 and continuous bubbling

with 95%/5% $O_2/CO_2$. Following recovery, slices were placed in a recording chamber and perfused with Krebs buffer at a rate of 3 ml/min and maintained at 34–36° C. For all experiments DNQX (10 µM), picrotoxin (100 µM), and CGP55845 (300 nM) were included in the solution to block AMPA, GABA-A, and GABA-B receptors. In all experiments except those involving L-DOPA, sumatriptan (1 µM) was also included in the bath. In all experiments, a gigaohm seal was made on a dopamine neuron in the subatantia nigra with a glass electrode (1.3–1.8 megaohm resistance) filled with an internal solution containing (in mM) 10 BAPTA (4 k), 90 K-methanesulphonate, 20 NaCl, 1.5 $MgCl_2$, 10 HEPES (K), 2 ATP, 0.2 GTP, and 10 phosphocreatine. In this configuration, the firing rate of the cell was measured. All cells were firing in a pacemaker fashion between 0.75 and 4 Hz and the firing frequencies were recorded. Then the seal was broken and whole-cell voltage clamp recordings (held at − 60 mV with an axopatch-1D amplifier) were achieved. The cell capacitance, input resistance, and series resistance were documented. $I_h$ currents were measured using a 60 mV hyperpolarization. D2-IPSC measurements were conducted in voltage clamp and continuously recorded and monitored using Chart 7 (AD Instruments, Colorado Springs, CO). To test for D2 receptor IPSCs, a glass electrode filled with Krebs buffer was lowered into the slice 20–50 µM away from the cell of interest. To begin, five 0.5 ms pulses at 40 Hz were applied with a stimulus isolator (World Precision Instruments) once every minute and recorded in an episodic manner with AxoGraph software (Berkeley, CA). The stimulation intensity was started at 1 µA. If no IPSC was produced, the stimulation was gradually increased and the stimulation electrode was repositioned. The stimulation intensity of 12 µA was never exceeded, as this would often cause loss of recording or unclamped depolarization. In some experiments, the stimulation was reduced to a single pulse. To measure cell excitability, recordings were done in current clamp. A current was injected such that the membrane potential was − 70 mV to quiet pacemaker firing, and current steps were applied in + 50 pA intervals (from + 50 − + 250 pA for 500 ms each) over the injected holding current, and the number of action potentials during each step was recorded. The application of drugs was done using bath superfusion in all experiments. For the L-DOPA experiments, 10 µM L-DOPA was applied for 10 min, and then removed for 10 min before applying sumatriptan (1 µM). For testing D2 receptor sensitivity, 1 µM dopamine was applied until a peak current was achieved and then removed. Once the current returned to a steady baseline, 100 µM dopamine was applied until a new peak was achieved. For amphetamine experiments, 10 or 30 µM were applied for 5 min (*Figure 2C and D*), followed by the application of sulpiride, or applied until the rise in current had nearly stopped and then quinpirole was added (*Figure 2—figure supplement 1*). All electrophysiological experiments were performed and analyzed by an experimenter blind to the genotype.

## Sholl analysis of individual dopamine neurons

Dopamine neurons for morphological analyses were first identified and characterized electrophysiologically. Whole cell recordings were made from one cell per slice using an internal solution that contained neurobiotin (0.05%). Cells were recorded for a minimum of 15 min prior to removal of the electrode, and capacitance and resistance were recorded (no differences were observed between RIM control and RIM cKO[DA] cells, not shown). Slices were then incubated for 30–45 min in extracellular solution at 35 °C prior to fixation. Slices were fixed for 30 min in PBS (phosphate buffered saline) + 4% paraformaldehyde (PFA) at room temperature and washed 3 × 15 min in PBS. Slices were incubated in PBS + 0.5% fish skin gelatin (FSG) + 0.3% Tween 20 with streptavidin Alexa Fluor 568 (at 1:1000) and washed 3 × 15 min in PBS at room temperature. Sections were mounted on glass slides with coverglasses with a 1.5 refractive index using fluorescent mounting medium. Laser-scanning confocal images were acquired on a ZEISS LSM880 with airyscan, with laser excitation at 560 nm at 20 x magnification with Z steps of 1 µm. The images were acquired through the Z plane such that the whole cell and all processes in the slice were captured. The Z-stacks were collapsed with z-project using Fiji for analysis. Individual neurons in confocal images were then traced manually. Sholl analysis of each traced neuron was performed using the ImageJ Sholl analysis plug-in. The cell body was selected, and the number of neurite crossings of concentric circles around the center of the cell body was measured at increasing radii (from 20 µm to 180 µm in 10 µm intervals). Cell filling, image acquisition and quantification were performed by experimenters blind to the genotype.

## Data analyses and statistics

To analyze IPSC amplitudes, the maximum amplitude for a given IPSC was recorded between 200 and 400 ms following the stimulation onset. This range was used because the stereotyped peak of D2-IPSCs reliably falls within it and a measurement for very small or non-existent IPSCs would still be generated. Each reported value is the average of three consecutive stimulation events. For analysis of spontaneous IPSCs, AxoGraph software was used to continuously collect data (sampling at 10 kHz) following the application of cocaine (300 nM) and forskolin (1 µM). For analysis, recordings were filtered at 1 kHz and decimated (averaging 10 points). Spontaneous IPSCs were automatically detected using a sliding template procedure in AxoGraph. The template was generated by averaging multiple events that conformed to previously published kinetic analysis (*Gantz et al., 2013*). Spontaneous IPSCs were only detected with amplitudes greater than 2.1 x the standard deviation of baseline noise. Detected events were manually examined for quality assurance. Statistics were performed using GraphPad Prism. All data are shown as mean ± SEM, and individual cells are shown as small circles. Student's t-test was used for comparison of two groups while ANOVA was used to compare more than two groups or if there were multiple variables. Tukey's (for one-way) or Sidak's (for two-way) multiple comparison tests were used if a repeated measures ANOVA reached significance and Bonferroni post-hoc comparisons were done if the one- or two-way ANOVA reached significance.

## Drugs

Drugs were bath applied. MK-801, forskolin, CGP55845, and picrotoxin were acquired from Hello Bio Princeton, NJ. CNQX, sulpiride, L-DOPA, were acquired from Sigma-Aldrich. Amphetamine was acquired from NIH NIDA and sumatriptan was acquired from Glaxo Wellcome Inc.

## Acknowledgements

This work was supported by the National Institutes of Health (R01NS083898 and R01NS103484 to PSK, R01DA04523 to JTW, K99DA044287 to BGR), by the Dean's Initiative Award for Innovation (to PSK), and by a Harvard-MIT Joint Research Grant (to PSK). We thank Dr. Changliang Liu for insightful discussions.

## Additional information

### Funding

| Funder | Grant reference number | Author |
| --- | --- | --- |
| National Institute of Neurological Disorders and Stroke | R01NS083898 | Pascal S Kaeser |
| National Institute on Drug Abuse | R01DA04523 | John T Williams |
| National Institute of Neurological Disorders and Stroke | R01NS103484 | Pascal S Kaeser |
| National Institute on Drug Abuse | K99DA044287 | Brooks G Robinson |
| Harvard Medical School | | Pascal S Kaeser |

The funders had no role in study design, data collection and interpretation, or the decision to submit the work for publication.

### Author contributions

Brooks G Robinson, Conceptualization, Formal analysis, Funding acquisition, Investigation, Methodology, Writing—original draft; Xintong Cai, Formal analysis, Investigation, Methodology, Writing—review and editing; Jiexin Wang, Resources, Investigation; James R Bunzow, Investigation, Writing—review and editing; John T Williams, Conceptualization, Formal analysis, Supervision, Funding acquisition, Investigation, Writing—original draft, Project administration; Pascal S Kaeser,

Conceptualization, Formal analysis, Supervision, Funding acquisition, Writing—original draft, Project administration

### Author ORCIDs
Brooks G Robinson (iD) https://orcid.org/0000-0001-5020-531X
John T Williams (iD) http://orcid.org/0000-0002-0647-6144
Pascal S Kaeser (iD) https://orcid.org/0000-0002-1558-1958

### Ethics

Animal experimentation: All animal experiments were performed according to institutional guidelines of Harvard University and of Oregon Health & Science University, and were in strict accordance with the recommendations in the Guide for the Care and Use of Laboratory Animals of the National Institutes of Health. The animals were handled according to protocols (protocol numbers Harvard IS00000049, OHSU IP00000160) approved by the institutional animal care and use committee (IACUC).

### Decision letter and Author response
Decision letter https://doi.org/10.7554/eLife.47972.011
Author response https://doi.org/10.7554/eLife.47972.012

## Additional files

### Supplementary files
• Transparent reporting form
DOI: https://doi.org/10.7554/eLife.47972.009

### Data availability
All data generated during this study are included in the figures with individual data points shown in each figure whenever possible.

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
