## [Decision Letter]

Thank you for submitting your article "RIM is essential for stimulated but not spontaneous somatodendritic dopamine release in the midbrain" for consideration by *eLife*. Your article has been reviewed by three peer reviewers, and the evaluation has been overseen by a Reviewing Editor and Eve Marder as the Senior Editor. The following individuals involved in review of your submission have agreed to reveal their identity: Nicolas Tritsch (Reviewer #1); Pablo E Castillo (Reviewer #2); Louis-Eric Trudeau (Reviewer #3).

The reviewers have discussed the reviews with one another and the Reviewing Editor has drafted this decision to help you prepare a revised submission.

Summary:

The study by Robinson et al. shows that RIM proteins are required for evoked somatodendritic release of dopamine from midbrain DA neurons in mice. Specifically, selective removal of RIM from dopamine midbrain DA neurons abolished D2 receptor mediated-IPSCs induced by electrical bulk stimulation, whereas spontaneous release of dopamine remained intact. Together the data indicate that the mechanisms underlying evoked and spontaneous release of dopamine differ.

Essential revisions:

1) Validation of DA neurons:

- What midbrain dopamine neurons were analyzed in this study (SN, VTA or both)?

- Are dopamine neurons in cKO mice just as likely to be stimulated electrically to release dopamine than control neurons? Please compare spontaneous firing rates and action potential waveforms, and then demonstrate that electrical stimulation equally recruits dopamine neurons in both genotypes.

- Is the global density of dopaminergic dendrites was altered in these KO mice (suggested experiment:TH immunostaining in the substantia nigra)?

2) D2-IPSCs: To more directly address the possibility of changes in D2 receptor responsiveness of dopamine neurons in the RIM KO mice, please measure membrane currents evoked in dopamine neurons by local application of a D2 agonist.

The kinetics of spontaneous D2-IPSCs and electrically evoked D2-IPSCs seem to differ as seen in Figure 3B. Please measure and analyze the frequency, rise-time and decay of evoked and spontaneous D2-IPSCs.

3) RIM-dependent synchronization:

Can action potentials triggered by direct depolarization of DA neurons induce DA-IPSCs? If so, the authors may want to use this single-cell approach to confirm that RIM-dependent synchronization is stimulation protocol-independent and can be observed with a more subtle manipulation.

4)Rim localization:

The authors need to state more explicitly what is known about the localization of RIM1-2 in neurons and cite the relevant work. Can the authors show the presence of RIM proteins in the somatodendritic domain of dopamine neurons? Can the authors detect RIM in dendrites (As: "In dopamine axons, RIM was only present in ~1/3 of the varicosities…")?

Reviewer #1:

This brief report makes a simple point, and does so clearly and well: it reveals that the presynaptic protein RIM is essential for electrically evoked, but not spontaneous dendritic release of dopamine in the mouse midbrain. This paper comes on the heels of a compelling report from the same last author, showing that the active zone proteins RIM 1 and 2 distribute to presynaptic terminals of midbrain dopaminergic neurons in striatum, and that mice in which these proteins are conditionally deleted (cKO) from dopaminergic neurons are unable to release dopamine when measured with amperometry (Liu et al., 2008). Interestingly, these mice did not appear to suffer from gross locomotor defects and microdialysis experiments revealed trace amounts of extracellular dopamine in striatum, suggesting that spontaneous release may be intact. Here, the authors build upon this work, comparing dendritic release of dopamine in of control and RIM cKO mice using whole cell recordings of D2 receptor-evoked GIRK currents in midbrain dopamine neurons. They report the absence of electrically-evoked GIRK currents in RIM cKO mice, and perform a series of control experiments to rule out insufficient stimulation, or changes in D2 receptor expression. Interestingly, spontaneously occurring D2 receptor mediated GIRK currents appear intact in these mice, harking back to their earlier speculation that spontaneous release may occur in a RIM-independent fashion.

I only have 2 points that should be clarified, or excluded as potential confounds to confirm their conclusions:

- In Figure 2, the authors use amphetamine to test whether the cKO mice have a reduced number of vesicles (as opposed to a defect in release). They report no differences in current amplitude compared to WT mice, but a difference may only be revealed using this approach if GIRK conductances are not saturated. Can the authors rule this out? A direct measure of large dense core vesicles with EM would obviously be a better experiment, but probably not a resource available to either lab.

- In Figure 1—figure supplement 1, the authors make the case that the intrinsic properties of dopamine neurons are identical in both genotypes. But what the authors really need to show is that dopamine neurons in cKO mice are just as likely to be stimulated electrically to release dopamine than control neurons. The authors could start by comparing spontaneous firing rates and action potential waveforms, and then demonstrate that their electrical stimulation paradigm equally recruits dopamine neurons in both genotypes.

Reviewer #2:

In this study Robinson et al. report that RIM proteins are required for evoked somatodendritic release of dopamine from midbrain DA neurons in mice. The key finding is that selective removal of RIM from these neurons abolished D2-IPSCs induced by electrical bulk stimulation, whereas spontaneous release of dopamine remained intact. These observations strongly suggest that the mechanisms underlying evoked and spontaneous release of dopamine differ. The need for spatiotemporal precision of dopamine release and the associated secretory machinery, as seen for fast neurotransmitters, is unclear to this reviewer. In any event, the experiments are well designed and the results support the claims that are made. I only have a few suggestions that may strengthen the authors' conclusions.

1) "RIMs scaffolding functions may allow for the synchronization of release from multiple sites or multiple vesicles". Synchronization may also result from the stimulation method (extracellular bulk stimulation). Can action potentials triggered by direct depolarization of DA neurons induce DA-IPSCs? If so, the authors may want to use this single-cell approach to confirm that RIM-dependent synchronization is stimulation protocol-independent and can be observed with a more subtle manipulation.

2) The comparison between single stimulus and five stimuli IPSCs can only provide a very indirect estimation of Pr. Yet, the authors conclude that evoked somatodendritic release of DA has a high initial Pr and that a scaffolding mechanism to tether dopamine-laden vesicles to calcium channels and other secretory machinery are essential. Repetitive, bulk stimulation could release modulators that reduce transmitter release that can mistakenly interpreted as high initial Pr.

3) Why does somatodendritic DA release require the spatiotemporal precision that is typically observed in fast neurotransmission? Evoked DA-IPSCs are extremely slow -as compared to fast synaptic transmission- and do not seem to require (pre-postsynaptic) nanodomains. This issue should be discussed more thoroughly.

4) "In dopamine axons, RIM was only present in ~1/3 of the varicosities…". Can the authors detect RIM in dendrites?

5) The authors claim that spontaneous D2-IPSCs were kinetically similar to electrically evoked D2-IPSCs. However, consistent with Gantz et al., 2013 their kinetics seem to differ as seen in Figure 3B. Maybe the authors want to measure and analyze the rise-time and decay of evoked and spontaneous D2-IPSCs.

Reviewer #3:

In this manuscript by Robinson and colleagues, the authors explored the implication of the active zone proteins RIM1-2 in somatodendritic dopamine release in mice. The work is quite original as the molecular mechanisms of somatodendritic dopamine release are still incompletely characterized and whether the release machinery and scaffolding proteins involved are similar or different from the machinery in axon terminals in an important question. Globally, the work presented in this very short paper is quite convincing. The block of dendritic dopamine release is quite striking. However, a few issues need to be addressed.

1) The authors need to state more explicitly what is known about the localization of RIM1-2 in neurons and cite the relevant work. Obviously, it would add significantly to the paper to demonstrate the presence of RIM proteins in the somatodendritic domain of dopamine neurons. The authors should at least explain why this was not done in the present work (difficulty to distinguish between RIM protein directly in the dendrite as opposed to RIM in terminals contacting the dendrites?)

2) The Introduction should make it clearer that somatodendritic dopamine release is well established to be vesicular in origin and to be sensitive to cleavage of SNARE protein by clostridial toxins.

3) The data on the frequency-dependence of dendritic dopamine release presented in Figure 1—figure supplements 2 and 3, while interesting, are not really relevant to the main point of the paper. Also, the conclusions drawn from these results (that dendritic dopamine release employs scaffolds and that RIM heterozygosity did not strongly impair dendritic dopamine release) are not really supported by this data. I would suggest removing this data from the manuscript.

4) The experiments carried out with L-DOPA are presented as if they were performed to determine whether the loss of dendritic dopamine release in RIM KO mice is due to reduced levels of dopamine D2 receptors. In fact, the results do not really address this question, which is otherwise important. The fact that L-DOPA induces a D2R-dependent response which apparently comes from serotonin axon terminals is quite interesting in itself, but only supports the idea that some dopamine D2 receptors remain in the somatodendritic compartment of dopamine neurons in the KO mice. It does not allow saying whether there are less or more D2R. Also, in control mice, it is surprising to note that there was no difference between the baseline response and the response after L-DOPA and sumatriptan. This would tend to suggest that L-DOPA did not boost dendritic dopamine release at all. Also, the sumatriptan-sensitive component is bigger in RIM KO mice compared to the control mice. What this means is unclear. Was there some compensatory adaptation in the 5-HT terminals in response to abrogation of axonal and dendritic dopamine release in these mice? Globally, these experiments with L-DOPA raise more questions than anything else.

5) To more directly address the possibility of changes in D2 receptor responsiveness of dopamine neurons in the RIM KO mice, it seems to me that it would be better to simply measure membrane currents evoked in dopamine neurons by local application of a D2 agonist. This would nicely complement the experiments of Figure 2 with amphetamine.

6) The experiments of Figure 3 are quite interesting. The authors should also provide the frequency of these spontaneous events in control and KO neurons.

7) The absence of RIM starting from the late embryonic period could have perturbed the development of dopamine neurons, potentially leading to altered dendritic development. The capacitance results from Figure 1—figure supplement 1 argue that there were no major changes in the size of dopamine neurons in the KO mice. However, do the authors have the result of a TH immunostaining experiment in the substantia nigra to determine whether the global density of dopaminergic dendrites was altered in these KO mice?

---

## [Author Response]

1) Validation of DA neurons:- What midbrain dopamine neurons were analyzed in this study (SN, VTA or both)?

We apologize that we did not make this clear. All analyzed neurons were dopamine neurons in the pars compacta of the substantia nigra. This is now mentioned at various places including the results, the legend of Figure 1, and the Materials and methods section.

- Are dopamine neurons in cKO mice just as likely to be stimulated electrically to release dopamine than control neurons? Please compare spontaneous firing rates and action potential waveforms, and then demonstrate that electrical stimulation equally recruits dopamine neurons in both genotypes.

We have performed the requested experiments. First, we have quantified spontaneous firing rates and there was no difference between RIM control and RIM cKO^DA^ neurons (in Figure 1—figure supplement 1). We have then performed experiments to assess whether stimulation recruits the RIM knockout neurons efficiently. We have quantified the number of action potential spikes induced by current injections in dopamine neurons of RIM control and RIM cKO^DA^ mice (new Figures 2C and 2D), and found that stimulation equally recruits dopamine neurons in both genotypes. The observations that pacemaker firing, firing in response to depolarization, and input resistance are unchanged strongly indicate that excitability is normal in RIM cKO^DA^ neurons.

An additional way to assess this would be to determine the somatic action potential waveforms of dopamine neurons. We have not attempted this here because 1) all recordings were done with amplifiers that cannot truly follow rapid voltage changes and hence action potential shape may be distorted (Magistretti et al., 1996) and 2) the spike and afterhyperpolarization of these neurons include many conductances, some of which would only cause subtle changes in the action potential waveform if affected. A recent review on dopamine neurons points out this complexity (Gantz et al., 2018). We strongly feel that any change could easily be missed and would prefer not to reach conclusions about the many important conductances based on simply assessing somatic waveforms. Nevertheless, the observations that spontaneous firing, recruitment upon depolarization and input resistance are unchanged exclude that excitability defects account for the strong impairment in release in RIM cKO^DA^.

- Is the global density of dopaminergic dendrites was altered in these KO mice (suggested experiment:TH immunostaining in the substantia nigra)?

We thank the reviewers for bringing this up and have performed experiments to address this point. First, we have stained TH neurons in RIM control and RIM cKO^DA^ mice in preliminary experiments. While the overall morphology and TH density appeared unaffected, dendrite and cell shapes could not be analyzed with certainty because the dendrites are dense and for many dendrites it is not possible to follow and assign them to a given cell.

Instead, we performed a full experiment in which we identify individual dopamine neurons electrophysiologically in acute brain slices and then dye-filled these neurons. We then fixed the slices and imaged individual dopamine neurons in RIM control and RIM cKO^DA^ slices by confocal microscopy, and performed Sholl-analyses (new Figures 2A and 2B) to analyze their branch structure. RIM knockout dopamine neurons appear identical to the control neurons and electrophysiological properties of the neurons were unchanged (data not shown as they would duplicate the data in Figure 1—figure supplement 1).

These data establish that there are no strong effects of RIM deletion on dopamine neuron cell size and shape.

2) D2-IPSCs: To more directly address the possibility of changes in D2 receptor responsiveness of dopamine neurons in the RIM KO mice, please measure membrane currents evoked in dopamine neurons by local application of a D2 agonist.

We thank the reviewers for bringing this up and have performed this experiment as instructed (Figure 1—figure supplement 3). We have measured D2 receptor currents in response to bath application of 1 µM followed by 100 µM dopamine. There is no significant difference in the currents induced by dopamine between RIM control and RIM cKO^DA^ mice. While a robust response is present in both genotypes upon application of 100 µM dopamine, there may be a non-significant trend towards a reduction in RIM cKO^DA^ mice in 1 µM dopamine. We note that these currents are small and unreliable. However, overall our data strongly support that D2 receptor localization and function is not significantly impaired, because the D2 responses to L-DOPA (metabolized and released by serotonin neurons, Figures 1C and 1D) and amphetamine (Figure 2—figure supplement 1) are unchanged, and most importantly the amplitude of spontaneous D2 IPSCs is unaffected by RIM cKO^DA^ (Figure 3).

The kinetics of spontaneous D2-IPSCs and electrically evoked D2-IPSCs seem to differ as seen in Figure 3B. Please measure and analyze the frequency, rise-time and decay of evoked and spontaneous D2-IPSCs.

We have performed new experiments and analyses of the frequency and kinetics of spontaneous events in RIM control and RIM cKO^DA^ animals. We note that these experiments are laborious given the low amplitude and overall frequency of the events and the difficulty of their detection. Our analysis reveals that the frequency of spontaneous events is not significantly changed in the RIM cKO^DA^ mice, but there is a sizable trend towards an increase in spontaneous event frequency (Figure 3C). We also provide a new analysis of the 20% peak width of the spontaneous events that establish that the kinetics of the spontaneous events are unchanged in RIM cKO^DA^ mice.

The kinetics of spontaneous and evoked D2-IPSCs and how they relate to one another have been extensively studied before (Courtney and Ford, 2014; Gantz et al., 2013). As presented by these studies and in the previous version of our manuscript, the tail of the evoked D2-IPSC is somewhat longer than that of the spontaneous D2-IPSC. It has also been shown in these previous studies that the spontaneous D2-IPSCs scale to evoked D2-IPSCs, and the decay is most likely dependent of downstream GPCR signaling with a decay time constant of 300-350 ms rather than the properties of dopamine release. While interesting, we strongly feel that a detailed analysis and discussion of these points is marginal to the manuscript and distracts from the main point. In the interest of keeping this brief report concise and focused on the key point (the mechanisms of somatodendritic dopamine release), we have removed the comparison of the spontaneous and evoked D2-IPSCs, but instead refer to the relevant literature.

3) RIM-dependent synchronization:Can action potentials triggered by direct depolarization of DA neurons induce DA-IPSCs? If so, the authors may want to use this single-cell approach to confirm that RIM-dependent synchronization is stimulation protocol-independent and can be observed with a more subtle manipulation.

We have pursued experiments to test whether direct depolarization of a DA neurons triggers DA IPSCs in that neuron in previous experiments. Unfortunately, we have never succeeded in using a single neuron as a “sensor” for its dopamine release. Hence, this experiment is not possible. We have adjusted the text around the hypothesis of synchronization and have included the possibility suggested by reviewer 3 (synchronization of inputs across neurons, main text, eleventh paragraph) to better express the hypothetical nature of these models and their underlying mechanisms.

4)Rim localization:The authors need to state more explicitly what is known about the localization of RIM1-2 in neurons and cite the relevant work. Can the authors show the presence of RIM proteins in the somatodendritic domain of dopamine neurons? Can the authors detect RIM in dendrites (As: "In dopamine axons, RIM was only present in ~1/3 of the varicosities…")?

We thank the reviewers to raise the point of RIM localization and we now address it (main text, fourth paragraph). RIM localization has been best characterized in hippocampal neurons. There, RIM is predominantly found in axons. Within axons, it is highly concentrated at active zones (Kaeser et al., 2011; Wang et al., 1997), as best assessed by STED microscopy (de Jong et al., 2018). Two reports, however, proposed that RIM also may have dendritic functions. Specifically, it was proposed that RIM1α may support postsynaptic trafficking in dendrites (Wang et al., 2018), and RIM3 and 4, small RIM version, localize in dendrites and control their growth (Alvarez-Baron et al., 2013).

In the dopamine neurons, RIM localization has previously only been assessed in striatal axons of substnatia nigra and VTA dopamine neurons, where it is clustered in sparse, active zone-like release sites (Liu et al., 2018). Recently, we have put significant effort into attempting to localize RIM in somata and dendrites of dopamine neurons with several super-resolution microscopic methods. Unfortunately, we have remained uncertain about RIM localization in these compartments. The key challenge is that dopamine somata and dendrites receive dense presynaptic inputs from other neurons, and hence somata and dendrites are densely decorated with RIM, but it is impossible to tell whether some (likely a very small fraction) of these RIM clusters are within somata and dendrites (as opposed to in active zones of nerve terminals onto somata and dendrites, only separated by a synaptic cleft of ~20 nm width from the inside of the dopaminergic cell). Hence, despite significant effort, we have not been able to establish or exclude somatic and dendritic localization of RIM at release-site like hotspots.

This point also relates to the broader question of whether there are defined release sites (for example generated or marked by RIM) in dopamine neuron somata and dendrites. While our data suggest that secretory hotspots in dopamine neurons are close to dopamine receptors on receiving cells, it has remained uncertain how this secretory pathway is organized. For example, fundamental aspects such as the nature of the membranous compartment that mediates somatodendritic release (small clear vesicles vs. tubulovesicle vs. other vesicles) has remained uncertain. In respect to RIM, it is possible that it is clustered at release sites, decorates the target membrane broadly, is instead associated with the vesicular compartment that contains dopamine, or is a soluble, cytosolic release factor to support release. While all data support the model that RIM is sparse in somata and dendrites, it is currently impossible to distinguish between these different models.

We hope that the better description of what is known about RIM localization in the text and the explanations provided here are sufficient to clarify this point. We will continue to study the structures that underlie somatodendritic dopamine exocytosis. We also hope that our findings motivate other laboratories to join the important effort to understand the structure and organization of the somatodendritic secretory pathway in dopamine neurons.

Reviewer #1:[…] I only have 2 points that should be clarified, or excluded as potential confounds to confirm their conclusions:- In Figure 2, the authors use amphetamine to test whether the cKO mice have a reduced number of vesicles (as opposed to a defect in release). They report no differences in current amplitude compared to WT mice, but a difference may only be revealed using this approach if GIRK conductances are not saturated. Can the authors rule this out? A direct measure of large dense core vesicles with EM would obviously be a better experiment, but probably not a resource available to either lab.

As outlined above in the response to the editorial decision, we have ruled out that D2 GIRK conductances are saturated (Figure 2—figure supplement 1).

We also agree that it would be great to have an electron microscopic assessment of the vesicular compartment for dopamine release in the RIM cKO^DA^ mice, but we note that despite many years of research it has not been established what vesicle dopamine is released from in wild type animals. We hope that the reviewer understands that we cannot solve this problem in the context of a revision.

- In Figure 1—figure supplement 1, the authors make the case that the intrinsic properties of dopamine neurons are identical in both genotypes. But what the authors really need to show is that dopamine neurons in cKO mice are just as likely to be stimulated electrically to release dopamine than control neurons. The authors could start by comparing spontaneous firing rates and action potential waveforms, and then demonstrate that their electrical stimulation paradigm equally recruits dopamine neurons in both genotypes.

We have now included these experiments as outlined in the response to the editorial decision letter (major point 1) in Figure 1—figure supplement 1 and in Figure 2.

Reviewer #2:[…] I only have a few suggestions that may strengthen the authors' conclusions.1) "RIMs scaffolding functions may allow for the synchronization of release from multiple sites or multiple vesicles". Synchronization may also result from the stimulation method (extracellular bulk stimulation). Can action potentials triggered by direct depolarization of DA neurons induce DA-IPSCs? If so, the authors may want to use this single-cell approach to confirm that RIM-dependent synchronization is stimulation protocol-independent and can be observed with a more subtle manipulation.

Unfortunately, the single cell approach has not been used successfully. In the past, this experiment has been taken on by many researchers in each of the monoamine nuclei, locus coeruleus, dorsal raphe, substantia nigra and VTA. Pharmacological experiments did further not find any evidence of autoreceptor dependent inhibition. Similarly, paired cell recordings have been the source of failed experiments in both the locus coeruleus and VTA. At this point, the site(s) of dendritic connections are not known. A similar experiment can be done in autaptic culture, but in this preparation the release is most likely axonal rather than dendritic. Finally, experiments using the intracellular application of reserpic acid, a membrane impermeant version of reserpine, did not block spontaneous IPSCs suggesting that at least the spontaneous events were not true autoreceptor dependent processes (Gantz et al., 2015). We hope that given these explanations, the reviewer agrees that this experiment is currently not possible.

Instead, we have toned down the Discussion and also express that synchronization may arise from synchronization of different neurons. We note that the RIM-dependent release machinery would also be well suited to do this as it time-locks release with firing. Hence, if dopamine neurons receive timed depolarizing inputs, the fast secretory machinery may serve to convey these signals as well.

2) The comparison between single stimulus and five stimuli IPSCs can only provide a very indirect estimation of Pr. Yet, the authors conclude that evoked somatodendritic release of DA has a high initial Pr and that a scaffolding mechanism to tether dopamine-laden vesicles to calcium channels and other secretory machinery are essential. Repetitive, bulk stimulation could release modulators that reduce transmitter release that can mistakenly interpreted as high initial Pr.

We agree to this point and thank the reviewer for pointing this out. Given this comment and the comments of the other reviewers, we have removed this figure. We further note that previous literature has established a high Pr (Beckstead et al., 2007), and we simply refer to this previous literature (which contains a much more extensive characterization than the one we provided in the previous version of the paper) in the interpretation of our data.

3) Why does somatodendritic DA release require the spatiotemporal precision that is typically observed in fast neurotransmission? Evoked DA-IPSCs are extremely slow -as compared to fast synaptic transmission- and do not seem to require (pre-postsynaptic) nanodomains. This issue should be discussed more thoroughly.

It has remained uncertain what the true time scales of dopamine coding are and why such fast machinery has been selected through evolution to mediate its release. Independent of the rapidity, it is important that D2 receptors require a relatively high concentration of dopamine for activation (Courtney and Ford, 2014; Ford et al., 2009), and the machinery we describe here likely accounts for this. Finally, we would note that secretory machinery like RIM may also be well suited for regulation. We now better discuss this in the manuscript with a specific statement referring to the need for fast signaling.

4) "In dopamine axons, RIM was only present in ~1/3 of the varicosities…". Can the authors detect RIM in dendrites?

We have addressed this question in the response to the editorial decision major point 4, we hope that this clarifies what is known at this point.

5) The authors claim that spontaneous D2-IPSCs were kinetically similar to electrically evoked D2-IPSCs. However, consistent with Gantz et al., 2013 their kinetics seem to differ as seen in Figure 3B. Maybe the authors want to measure and analyze the rise-time and decay of evoked and spontaneous D2-IPSCs.

We thank the reviewer for this comment and have addressed it by removing the claim of kinetics. As explained above and as noted by the reviewer, this point has been addressed in previous literature, and we refer to this body of literature now.

More generally, the differences between evoked and spontaneous transmitter release are subject of considerable discussion (see Kaeser and Regehr, 2014 for an in depth discussion of the topic) and the dopamine neurons are not alone. The Gantz et al., 2013 paper did find a difference in the time course of the IPSCs in that the evoked IPSCs had a somewhat longer tail. This was observed in experiments with minimal stimulation, and stimulation was most often done near the cell soma. However, the location of the site(s) of spontaneous or evoked release are not known and could account for the difference, but in the end the reason is not known.

Reviewer #3:[…] A few issues need to be addressed.1) The authors need to state more explicitly what is known about the localization of RIM1-2 in neurons and cite the relevant work. Obviously, it would add significantly to the paper to demonstrate the presence of RIM proteins in the somatodendritic domain of dopamine neurons. The authors should at least explain why this was not done in the present work (difficulty to distinguish between RIM protein directly in the dendrite as opposed to RIM in terminals contacting the dendrites?)

We have addressed this point in detail in the response to the editorial letter, major point 4.

2) The Introduction should make it clearer that somatodendritic dopamine release is well established to be vesicular in origin and to be sensitive to cleavage of SNARE protein by clostridial toxins.

We have addressed this in the Introduction and cite the relevant literature.

3) The data on the frequency-dependence of dendritic dopamine release presented in Figure 1—figure supplements 2 and 3, while interesting, are not really relevant to the main point of the paper. Also, the conclusions drawn from these results (that dendritic dopamine release employs scaffolds and that RIM heterozygosity did not strongly impair dendritic dopamine release) are not really supported by this data. I would suggest removing this data from the manuscript.

We thank the reviewer for bringing this up and fully agree, we have removed these data as also explained in the response to the editorial letter.

4) The experiments carried out with L-DOPA are presented as if they were performed to determine whether the loss of dendritic dopamine release in RIM KO mice is due to reduced levels of dopamine D2 receptors. In fact, the results do not really address this question, which is otherwise important. The fact that L-DOPA induces a D2R-dependent response which apparently comes from serotonin axon terminals is quite interesting in itself, but only supports the idea that some dopamine D2 receptors remain in the somatodendritic compartment of dopamine neurons in the KO mice. It does not allow saying whether there are less or more D2R. Also, in control mice, it is surprising to note that there was no difference between the baseline response and the response after L-DOPA and sumatriptan. This would tend to suggest that L-DOPA did not boost dendritic dopamine release at all. Also, the sumatriptan-sensitive component is bigger in RIM KO mice compared to the control mice. What this means is unclear. Was there some compensatory adaptation in the 5-HT terminals in response to abrogation of axonal and dendritic dopamine release in these mice? Globally, these experiments with L-DOPA raise more questions than anything else.

We fully agree that the L-DOPA experiments do not indicate that the D2 receptor expression is identical in the two genotypes, and we currently do not have an approach to count D2 receptor numbers. We think that the data are important and would prefer to leave them in the manuscript because they show that RIM cKO^DA^ neurons have D2 dopamine receptors to sense dopamine. The striking absence of an IPSC cannot be explained by the complete or near complete loss of receptors.

The release of dopamine from 5-HT terminals and the augmentation of dopamine release from dopamine neurons has been studied in wild type animals before (Beckstead et al., 2004; Gantz et al., 2015). Those results show that the major increase in the IPSC results from the 5-HT terminals. There is also an increase in dendritic dopamine release that is relatively small compared to that from the 5-HT terminals. The new results in the control animals (which are heterozygous for RIM) mirror experiments in these wild type animals, and we hope that the reviewer concurs that there is value in having this comparison in the paper given that it was not possible to have wild type littermates as controls included. To us, the observation that sumatriptan completely blocked the IPSCs in the RIM cKO^DA^ animals was a gratifying confirmation of the severity of loss of dopamine exocytosis in these animals!

5) To more directly address the possibility of changes in D2 receptor responsiveness of dopamine neurons in the RIM KO mice, it seems to me that it would be better to simply measure membrane currents evoked in dopamine neurons by local application of a D2 agonist. This would nicely complement the experiments of Figure 2 with amphetamine.

We thank the reviewer for suggesting this experiment. We have performed it and included it in the revised manuscript (Figure 1—figure supplement 3). We have addressed the point in detail in the response to the editorial decision (major point 2) and also note that the spontaneous event amplitude does not change. Together, these observations make a strong effect on dopamine receptor localization or function very unlikely.

6) The experiments of Figure 3 are quite interesting. The authors should also provide the frequency of these spontaneous events in control and KO neurons.

We thank the reviewer for proposing this experiment. It is a laborious experiment given the low frequency and difficult detection, but we think that the new data added in response to this suggestion make the manuscript significantly stronger. This point is also addressed in the response to the editorial decision, major point 2.

7) The absence of RIM starting from the late embryonic period could have perturbed the development of dopamine neurons, potentially leading to altered dendritic development. The capacitance results from Figure 1—figure supplement 1 argue that there were no major changes in the size of dopamine neurons in the KO mice. However, do the authors have the result of a TH immunostaining experiment in the substantia nigra to determine whether the global density of dopaminergic dendrites was altered in these KO mice?

We have addressed this point with new experiments and described in the response to the editorial decision letter major point 1, and the new data are shown in Figures 2A and 2B. There is no detectable effect on dendritic development.